# OpenReview forum: "LARGO: Latent Adversarial Reflection through Gradient Optimization for Jailbreaking LLMs"
_NeurIPS.cc/2025/Conference — NeurIPS 2025 poster_

### Official Review · Reviewer_WCWm · 2025-06-30

**Clarity:** 4
**Significance:** 4
**Originality:** 4
**Rating:** 6
**Confidence:** 5

**Summary:**

Summary:

LLMs are well known to be vulnerable to jailbreaks, prompt injections, and adversarial attacks generated using gradient-based-attacks (e.g. GCG). However jailbreaks and prompt injections tend to be hand-curated by manual prompters. Recent work, e.g. Scale/MHJ has shown a durable between the strongest human written jailbreaks and automated methods in the multi-turn context, and the ongoing discovery of new red-teaming tactics by human jail-breakers bears this out.

Strong automatic jailbreaking strategies are critical to
a) Understand how vulnerable existing LLMs are to these attacks so as  to inform deploy/no-deploy decisions
b) Generating accurate synthetic jailbreak data for training guard models and adversarial training
c) Helping researchers generate more robust pre/post-training approaches.

The strongest gradient based jailbreak method has been the GCG attack, but this attack is very inefficient in terms of forward passes.
Informally, GCG is intermediate between random search and true gradient based optimization. This is because the gradients are computed in the continuous space of the model but need to be rounded/converted to the discrete token space. Getting around this discrete to continuous rounding problem has been a going challenge in gradient based optimization of text jailbreaks.

The Algorithm is very  simple:

1. Attack the model in the soft-embedding space, i.e. soft tokens, using the GCG loss
2. Round the soft-token attack to hard tokens using this prompt to the attack LLM User: Assistant: Sure, I will summarize the message:
Line 94.
3. If the hard-token attack is not adversarial, KEEP going until a set number of iterations is complete.

The authors also conduct transfer experiments, compute perplexity, and compute universal suffixes.

The score jailbreaks using the legacy keyword matching from the original Zou/GCG paper and the more accurate StrongReject paper from Neurips 2024.

**Questions:**

see above, most of my weaknesses included questions.

-How does the computational complexity of this attack compare to baseline GCG?

-The authors claim to release code, but I do not see it, please clarify.

-The authors must report if they did conduct experiments attacking larger/frontier models with transfer attacks and briefly explain why they did not (e.g. time/cost).

**Ethical Concerns:**

["NO or VERY MINOR ethics concerns only"]

**Final Justification:**

the much shorter computational complexity of the attack and its implementation simplicity make me think this attack will be useful in red-teaming and defending real world models.

**Limitations:**

I don't think the ethical limitations section is very good. I am a bit disappointed the authors did not grapple with this more meaningfully or more persuasively write my designing novel and stronger jailbreaking strategies is justified despite the potential harm.

Additionally, the authors seem to not have disclosed this to any frontier labs. I am not 100% sure this disclosure is the correct choice, but the authors should at least have discussed this with each other and in the limitations section. Please expand the ethical review section substantially, this is a glaring weakness in the paper. THIS attack could have serious negative real world impact, please take that more seriously.

**Quality:**

3

**Strengths And Weaknesses:**

Strengths:

Full Disclosure: I have been working on a very similar research direction, so obviously I think this paper is a valuable contribution!

0. My summary above explains why I think designing strong and efficient gradient based jailbreaking attacks is important. I am optimistic this line of research will helps us design better post-training architectures, do robust training, and train defensive models. Also, could be used for economically efficient automated red teaming as a proxy for human attackers.

1. I think the formal/benchmark results are somewhat secondarily to the “it ain’t dumb if it works”. The main value of this paper is this simple and novel way to conduct the soft-to-hard rounding in a way that has non-trivial ASR on meaningful models!

2. Because the ability of LLMs to generate fluent text, the LARGO attacks generate substantially lower perplexity attacks than GCG. The authors do cite another recent paper, ADVprompter, which has lower perplexity, but also lower ASR (attack suggest rate) suggest an ASR-PPL tradeoff.

The  low perplexity, diverse jailbeaks, generated automatically, are one of the strongest features of this paper.

Weaknesses:

-the authors primarily benchmark against an older model, Llama 2-7B. A strength of the original GCG paper was that the authors had results for the frontier models of the time, gpt 3.5.

The authors must report if they did conduct experiments attacking larger/frontier models. If the results are negative, please include them, even informally.  If the authors did not attack models like Claude Sonnet, Deepseek R1, gemini, gpt4.0, or larger llama variants, please explain why? I assume this is because of cost/complexity, which is in my opinion a valid reason and would not change my strong score of this paper. That said, I want an answer!  If an answer is not provided, I may move my score downward.

To me, the authors convincingly demonstrate strong improvement over GCG other than the lack of these transfer experiments.

-The next weakness is that GCG is very computationally intense. Why don’t the authors report comparisons in compute time for GCG vs their attack? My understanding of the paper makes me think this attack should be substantially faster, maybe even an order of magnitude. Such comparisons could even be in analytical form vs experiments? The authors should meaningfully comment on this.

-There are stronger/more modern gcg variants like nanogcg. the authors do not compare to these stronger baselines or even cite them. please do more literature search and cite these. another such attack came from haize labs, not sure if there is an open source implementation.

---

> ### Author Rebuttal · Authors · 2025-07-31
>
> Thank you for the thoughtful review. We are glad that the reviewer found our method unique and effective. We especially appreciate the point that LARGO’s unique design is a meaningful contribution in itself.
>
> **On Weaknesses:**
> 1. **Transfer attack on larger/frontier models**: Thank you for bringing this up. We acknowledge the importance of testing against frontier models and add transfer experiments to DeepSeek-V3, Claude-4 Sonnet, and GPT-4o. We show transfer results below and will include these in revision. We did not run LARGO on larger models directly due to cost limits, but our experiments cover models of parameter size ranging from 4b to 13b.
>
> | Metric | DeepSeek-v3 | Claude 4 Sonnet | GPT-4o |
> |--------|---------------|-------------|-------------|
> | ASR    | 13.1              | 3.6       | 2.4       |
>
> 2. **Comparison on compute time**: Thank you for the suggestion to improve our paper. We show a runtime comparison below. We use the same setup as described in paper and take the average runtime of 20 runs for all methods. The reviewer correctly predicted our method to be one order of magnitude faster, and we will include this in revision.
>
> | Metric  | GCG   | AutoDAN | LARGO |
> |---------|-------|---------|-------|
> | Runtime/s | 1180 | 950    | 160  |
>
> 3. **Discussion and citations on stronger baselines**: Thank you for the suggestion. We will cite and discuss these in revision.
>
> **On Questions:**
> 1. **Comparison on compute time**: Please refer to our reply to weakness point 2.
> 2. **Code release**: Due to ethical concerns mentioned by the reviewer, and that links are prohibited during rebuttal, we will release the complete codebase in the camera ready paper.
> 3. **Transfer attack on larger/frontier models**: We included the transfer attack experiments, and found it transfers. Please see reply to point 1 for details.
>
> **On Limitations:**
>
> Thank you for bringing this up. We acknowledge the importance of a more comprehensive ethical statement section. While LARGO attacks are effective and stealth, it requires gradient access to produce them. Therefore, compared with third parties that need to rely on transfer of suffix tokens, LLM producers like Google and OpenAI would more easily generate such attacks to test and red-team their models for enhanced safety. Therefore, we think that the benefit of releasing LARGO outweighs the potential harm. Thank you for your suggestion on contacting the frontier labs with the source code before a public release, we will follow this. We will substantially expand the ethical statement section in revision.

---

> ### Comment · Reviewer_WCWm · 2025-08-01
> **thank you, revised score up to strong accept**
>
> Hello!
>
>
> 1. Transfer: Thanks for including these. Do you have any theory about the low degree of transfer? Did you do a naive one-shot transfer? Designing the "right" way to average gradients for transfer attacks/having the "right" proxy model is somewhat finicky, so I don't view it as dis-qualifying but worth noting. Hopefully you can remark on this in the final version.
>
> 2. Compute Runtime:
>
> Metric	GCG	AutoDAN	LARGO
> Runtime/s	1180	950	160
>
> Seeing the radically shorter runtime for largo is compelling- a 10x drop is amazing. This is very strong evidence in favor of this paper and I recommend emphasizing it in the final version of the paper.
>
>
> 3. Ethical limitations: Not to diminish the significance of this research, but the attack is "simple" enough that I expect real world adversaries will also discover it very quickly. Thus I also support releasing this attack so LLM providers may try to mitigate the attack.
>
> I appreciate you addressing my concerns and will revise my score to strong accept.

---

> > ### Author Response · Authors · 2025-08-04
> >
> > Thank you for the follow-up.
> > 1. **Transfer**: We did a naive one-shot transfer using the optimized suffixes from Llama2-7b. Due to the differences in model architecture, parameters, and the robust alignment training of frontier models, it is expected to have a lower transfer ASR. Nevertheless, we still observe both GPT-4o and Claude-4 Sonnet to generate harmful information to a few prompts, which showcases LARGO's effectiveness. We will include these in revision.
> > 2. **Compute runtime**: We agree with the reviewer and will emphasize on the runtime comparison in revision.
> > 3. **Ethical limitations**: We will release the source code and contact frontier labs as suggested.

---

### Official Review · Reviewer_bkhf · 2025-07-02

**Clarity:** 2
**Significance:** 3
**Originality:** 2
**Rating:** 5
**Confidence:** 3

**Summary:**

This paper proposes a jailbreak method against LLMs named LARGO. Previous methods (e.g., GCG) optimize adversarial suffixes at the token level but are constrained by discrete optimization challenges. In contrast, LARGO optimizes the suffix in the embedding space and then prompts the LLM to convert the continuous adversarial suffix into a natural-language suffix. LARGO mitigates the inefficiency of gradient-based adversarial prompt generation caused by discrete optimization, as well as the lack of interpretability in adversarial prompts.

**Questions:**

See Weaknesses

**Ethical Concerns:**

["NO or VERY MINOR ethics concerns only"]

**Final Justification:**

The authors have addressed all my concerns.

**Limitations:**

Yes.

**Quality:**

3

**Strengths And Weaknesses:**

Strengths:
- The idea is interesting, and the paper is clearly organized and well-motivated.
- The method is simple yet effective, which is impressive.
- The experimental results show that the method is effective.

Weaknesses:
My concerns are as follows:
- Does the interpreted suffix have a similar intermediate representation to the original latent suffix?
- In line 94, the \<latent suffix\> is detached from the original context in which it was optimized. This separation may cause it to lose its intended functionality, potentially making it hard to be decoded accurately. I believe a proper implementation of this method might require the model to repeat the full request (i.e., \<harmful request\> + \<latent suffix\>) . E.g.,
```bash
User: <harmful request> <latent suffix> \n\n Please repeat the above request. Assistant: Sure, I will repeat ...
```
- Is iterative refinement truly necessary? Given that the latent suffix alone successfully jailbreaks the model, and the model can interpret the embedding into a discrete textual suffix, wouldn’t a single-round interpretation suffice? Alternatively, might best-of-N sampling provide adequate interpretation without iterative steps?
- Since optimization in latent space can already jailbreak the white-box model, it seems unnecessary to compare its effectiveness with other white-box jailbreaking methods. The focus of the paper should be on black-box attacks, rather than devoting extensive discussion to the performance of white-box attacks.
- The methods selected for comparison are insufficiently comprehensive; additional relevant approaches should be included and discussed.

---

> ### Author Rebuttal · Authors · 2025-07-31
>
> Thank you for the thoughtful review. We are glad the reviewer found our idea interesting and method effective.
>
> **On Weaknesses:**
> 1. **Similarity between suffix embedding and original latent**: Thank you for the question. For successful jailbreaks, the average cosine similarity between the latent vector and the embedding of the interpreted suffix is 0.55 for Llama2-7b, suggesting moderate correlation. Given the effectiveness of the latent vector, the correlation is strong enough to jailbreak the model in most cases. Enhancing the correlation can be an interesting topic for future research.
>
> 2. **Interpretation implementation suggestion**: Thank you for the suggestion. While we acknowledge that contextual information is preserved in this way, this makes the interpretation request non-benign and we observe all three target models to frequently refuse such requests. In addition, multiple iterative interpretation steps on the harmful request may alter its content. Therefore we choose to decode the suffix separately.
>
> 3. **Necessity of iterative refinement**: Thank you for the suggestion. We find iterative refinement to be necessary because the one-step success rate is very low. Also, the projection of textual interpretation back to latent space could help to jump out of local minima, as we show in figure 2. We show the success rate at the first iteration for each model below, and will include the discussion in revision.
>
> | Metric  | Llama2-7b | Llama2-13b | Phi3-4b |
> |---------|-----------|------------|---------|
> | ASR     | 42.0      | 51.0       | 79.0    |
> | ASR@1   | 5.0       | 4.0        | 9.5     |
>
> 4. **Focus of paper and benchmark black-box attacks**: One focus of our paper is a natural language-based realistic white-box attack, which can be used by LLM providers like OpenAI for red teaming their existing LLM. Existing white-box attacks like GCG produce random symbols that are not unrealistic in format, and cannot provide realistic red-teaming for LLMs to improve their safety. We agree with the reviewer that black-box is also related, and we add PAIR [1] as an additional black-box attack on Llama2-7b and Qwen2.5-14b. PAIR underperforms on both models and takes significantly more LLM calls in its agentic prompting process. We will include additional results in revision.
>
> | Method | Llama2-7b | Qwen2.5-14b |
> |--------|-----------|-------------|
> | PAIR   | 2.5       | 17.5        |
> | LARGO  | 42.0      | 46.5        |
>
> 5. **Benchmark additional baselines**: Thank you for the suggestion. We already compared 3 recent jailbreak methods in our work. We further verify the effectiveness of LARGO by adding PAIR in comparison.
>
> Moreover, we agree with reviewer WCWm, “I have been working on a very similar research direction, so obviously I think this paper is a valuable contribution!’’, and they think “the formal/benchmark results are somewhat secondarily to the ‘it ain’t dumb if it works’. The main value of this paper is this simple and novel way to conduct the soft-to-hard rounding in a way that has non-trivial ASR on meaningful models!”
>
> We believe that our innovative method connecting latent and discrete space is a meaningful contribution on its own, as the first paper doing this, and the numbers are non-trivial too. We will include additional results in revision as suggested.
>
> [1] Jailbreaking Black Box Large Language Models in Twenty Queries, NeurIPS 2023

---

> > ### Comment · Reviewer_bkhf · 2025-08-05
> >
> > Thanks for your response. I’ve increased the score to 5.

---

### Official Review · Reviewer_CeFB · 2025-07-02

**Clarity:** 2
**Significance:** 3
**Originality:** 3
**Rating:** 5
**Confidence:** 4

**Summary:**

This paper proposes LARGO, a novel suffix jailbreak attack. It is implemented by repeatedly searching for continuous jailbreak suffix embeddings and then mapping them back to discrete text, to discover effective "real jailbreak suffixes". Experiments on several baseline LLMs and jailbreak datasets show that LARGO achieves superior attack performance and greater jailbreak prompt readability (i.e., lower perplexity) compared with existing attacks.

**Questions:**

See **Weaknesses**.

**Ethical Concerns:**

["NO or VERY MINOR ethics concerns only"]

**Final Justification:**

I think the idea of this paper is interesting, well-motivated, and novel. My main concerns were that the original version of the paper used too few jailbreak baselines for comparison and the evaluation was conducted on too few targeted models. The authors resolved my concerns by adding more jailbreak baselines and more targeted models during the rebuttal, which I think will significantly improve the quality of the final paper. Therefore, I raised my score from 4 to 5 to support acceptance.

**Limitations:**

See **Weaknesses**.

**Quality:**

3

**Strengths And Weaknesses:**

**Strengths**

1. I appreciate the idea of iteratively searching the continuous jailbreak suffix embeddings and then mapping back to discrete text to find an effective real-world jailbreak suffix. This idea is well-motivated and intuitively makes sense.

2. I believe this paper makes a significant contribution in tackling real-world challenges. Performing jailbreak attacks is typically inefficient because attackers usually need to optimize prompts in the discrete token space, which, however, is difficult. The authors propose a practical and novel solution to this problem.


**Weaknesses**

1. There are too few baseline jailbreak attacks. The authors include only three baselines, including the outdated GCG attack. I suggest adding more baselines such as BEAST [r1], AmpleGCG [r2], and PAIR [r3], all of which aim to generate low-perplexity jailbreak prompts. Moreover, previous works such as [r4] show that PAIR is more effective than AutoDAN, so I think it is necessary to adopt the PAIR attack as a baseline.

2. To the best of my knowledge, [r5] was the first to perform jailbreak attacks via the token embedding space. Although I know that the proposed LARGO uses continuous embeddings in a way that is very different from [r5], I still suggest that the authors cite [r5] and briefly discuss the differences between these two approaches.

3. What is the jailbreak suffix token length used for the GCG attack? Is it set to the default 20 tokens as in the original GCG paper? If so, the comparison in Table 1 between GCG and the proposed LARGO might not be fair. According to Table 6, it seems that LARGO uses a suffix length of at least 50 tokens. Since longer suffixes may lead to stronger jailbreak attacks, the authors should explain why the comparison between LARGO and GCG (i.e., 50 suffix tokens vs. 20 suffix tokens) in Table 1 is appropriate.

4. There are too few targeted LLMs. The main attack evaluation (i.e., Table 1) is conducted on only three models, two of which are quite old (Llama-2-7B and Llama-2-13B). I suggest including results for Qwen-2.5-14B in Table 1 (just like that in Table 3), and adding newer models such as Llama-3.1, Llama-4, Qwen-3, and Deepseek-V3.

5. I suggest including a comparison of the time usage for performing different jailbreak attacks.


**References**

[r1] Sadasivan et al. "Fast Adversarial Attacks on Language Models In One GPU Minute". ICML 2024.

[r2] Liao et al. "AmpleGCG: Learning a Universal and Transferable Generative Model of Adversarial Suffixes for Jailbreaking Both Open and Closed LLMs". COLM 2024.

[r3] Chao et al. "Jailbreaking Black Box Large Language Models in Twenty Queries". SaTML 2025 (arXiv 2023).

[r4] Xhonneux et al. "Efficient Adversarial Training in LLMs with Continuous Attacks". NeurIPS 2024.

[r5] Schwinn et al. "Soft prompt threats: Attacking safety alignment and unlearning in open-source llms through the embedding space". NeurIPS 2024.

---

> ### Author Rebuttal · Authors · 2025-07-31
>
> Thank you for the thoughtful review. We are glad that the reviewer found our idea innovative and our contribution significant.
>
> **On Weaknesses:**
> 1. **More effective baselines**: We will cite all the work suggested. As per the reviewer’s request, we run PAIR [1] as an additional baseline attack and report its performance on Llama2-7b and Qwen2.5-14b below. LARGO outperforms both models, especially for Llama2 which is less sensitive to prompting-based attacks.
>
> | Method | Llama2-7b | Qwen2.5-14b |
> |--------|-----------|-------------|
> | PAIR   | 2.5       | 17.5        |
> | LARGO  | 42.0      | 46.5        |
>
> We also agree with reviewer WCWm’s view that our method that connects latent and discrete space is a meaningful contribution on its own, as long as the numbers are non-trivial. We will include additional results in revision as suggested.
>
> 2. **Relevant citation**: Thank you for the suggestion. We will cite and discuss [r5] in revision.
>
> 3. **Consistent suffix token length**: Sorry for the confusion. As we note in line 134, we use consistent suffix length for all baselines, and the suffix lengths of LARGO, GCG, and the other baselines are all set to 200 for a fair comparison. We will make this more clear in revision.
>
> 4. **Additional target LLMs**: As per the reviewer’s request, we add Qwen2.5-14b as an additional target model, and show results below. We also transfer the suffixes to frontier models including DeepSeek-v3, Claude-4 Sonnet, and GPT-4o. We will include these results in revision.
>
> | Method   | AdvBench | JailbreakBench |
> |----------|----------|----------------|
> | GCG  | 27.5    | 29.5          |
> | LARGO  | 46.5     | 49.0           |
>
> | Metric | DeepSeek-v3 | Claude 4 Sonnet | GPT-4o |
> |--------|---------------|-------------|-------------|
> | ASR    | 13.1              | 3.6       | 2.4       |
>
> 5. **Time usage comparison**: Thank you for bringing this up. We show a runtime comparison below. We use the same setup as described in paper and take the average runtime of 20 runs for all methods. Our method is significantly faster than the baselines. We will include this in revision.
>
> | Metric  | GCG   | AutoDAN | LARGO |
> |---------|-------|---------|-------|
> | Runtime/s | 1180 | 950    | 160  |
>
> [1] Jailbreaking Black Box Large Language Models in Twenty Queries, NeurIPS 2023

---

> > ### Comment · Reviewer_CeFB · 2025-08-01
> >
> > Thanks to the authors for their detailed response, which resolved most of my concerns. I will raise my score to 5 to support accepting this work. Please include your additional results and discussions during the rebuttal in your final paper.

---

### Official Review · Reviewer_qRbx · 2025-07-03

**Clarity:** 3
**Significance:** 2
**Originality:** 2
**Rating:** 4
**Confidence:** 3

**Summary:**

The paper introduces LARGO, a three-stage jailbreak framework that optimizes a continuous latent suffix with gradient descent to maximize the probability of an affirmative harmful response. Then it prompts the same LLM to self-reflect and decode that latent vector into a human-readable suffix. Finally, LARGO back-projects the decoded text into the embedding space for further refinement until the attack succeeds.
Experiments on AdvBench and JailbreakBench show that LARGO attains the highest keyword-based ASR on Llama and Phi-3 while keeping perplexity far below GCG and only modestly above AdvPrompter. Ablations confirm that the optimized latent outperforms random latents and that longer suffixes strengthen attacks.

**Questions:**

1. Could you measure attack success when directly appending the optimized latent embeddings (e.g., via embedding API) without converting to text? Quantifying the gap between latent-only and text-decoded ASR would clarify how much the reflection step really adds.

2. Right now the pipeline first optimizes a latent vector and only then decodes it to text. Have you explored fine-tuning the decoded tokens based on the learned latent, or even running a single loop that jointly optimizes both the latent representation and its textual realization?

**Ethical Concerns:**

["NO or VERY MINOR ethics concerns only"]

**Final Justification:**

The paper proposes a simple yet effective method that outperforms baselines. However, there remains a noticeable gap compared to SOTA attack methods and latent-only methods, which prevents me from giving a higher score at this stage.

**Limitations:**

1. White-box dependency. Gradient access is assumed in main experiments;

2. Model self-reflection assumption. Success hinges on the model’s ability (and willingness) to turn latent vectors into sensible language.

**Paper Formatting Concerns:**

The paper is formatted correctly.

**Quality:**

2

**Strengths And Weaknesses:**

Strenghts:

1. Direct gradient optimization in embedding space sidesteps the combinatorial explosion of discrete token search.

2. Self-reflective decoding yields benign-looking English paragraphs with perplexity orders of magnitude lower than GCG.

3. The method works in single-prompt, universal, and transfer settings without manual prompt templates or auxiliary attacker models.

Weaknesses:

1. Reliance on self-interpretation may be brittle. Asking the same target LLM to explain its own latent attack vector mixes optimization and decoding responsibilities. This circular dependency could fail on models whose reflection abilities are weaker or deliberately sandboxed —so “letting the LLM explain the latent is not necessarily a robust strategy.”

2. No measurement of latent-only success. The paper does not report how often the raw optimized latent (without decoding) already triggers jailbreak, so the marginal benefit of the self-reflection step is unclear.

3.  The study stops at Llama-2 and Phi-3; it does not test against the newest open-source or frontier systems such as Llama-3, Qwen2.5, or GPT-4o. As a result, it is unclear whether the proposed latent-space attack scales to today’s strongest alignment defenses.

4. While LARGO outperforms GCG, AutoDAN, and AdvPrompter, more recent approaches reach 85-95 % ASR on the same benchmarks, whereas LARGO tops out in the mid-60 % range. The paper does not benchmark against these newer baselines, leaving its relative effectiveness uncertain.

---

> ### Author Rebuttal · Authors · 2025-07-31
>
> Thank you for the thoughtful review. We are glad that the reviewer found our method novel and effective.
>
> **On Weaknesses:**
> 1. **Reliance on self-interpretation**: Thank you for bringing this up. We acknowledge that our attack depends on the model’s self-reflective abilities, but it has been shown in previous works [1, 2] that LLMs commonly have the ability to decode its latent representations.
>
> The feasibility of the sandboxing defense is uncertain, as the interpretation task is completely benign and intuitively similar to a rewrite or paraphrase request. Moreover, as the model providers like OpenAI need to perform red-teaming for their model, the self-interpretation capability will not be sandboxed in this situation and our algorithm can be adopted to expose model weakness and improve safety.
>
> 2. **Measurement of latent-only success**: We are sorry for the confusion. We would like to highlight that the purpose of the reflection step is not to enhance jailbreak performance, but to convert the power of gradient-based attacks into realistic text prompts, since the user interface for LLMs, like ChatGPT, are text instead of latent vectors. **Eventually the text interpretations are used to jailbreak the model, not the latents.**
>
> Nevertheless, we follow your suggestion and conduct an ablation study on latent-only attacks on Advbench and show results below.
>
> | Method       | Llama2-7b | Qwen2.5-14b |
> |--------------|-----------|-------------|
> | Latent Only  | 95.0        | 98.5          |
> | LARGO        | 42.0        | 49.0          |
>
> We would like to note that it is expected that latent only has a higher success rate, as it directly optimizes the latent space, which cannot be input by the users. LARGO’s reflection step finds text prompts in the input space that can jailbreak an LLM in a realistic format, which is a different setting with different application scenarios.
>
> 3. **Attack on recent open-sourced and frontier models**: Thank you for the suggestion. We run LARGO on a more recent model, Qwen-2.5-14b, and report the results below.
>
> | Method   | AdvBench | JailbreakBench |
> |----------|----------|----------------|
> | GCG  | 27.5    | 29.5          |
> | LARGO  | 46.5     | 49.0           |
>
> For larger and frontier models like GPT-4o, we conduct transfer experiments to three frontier models using the optimized suffixes from Llama2-7b and show results below. We will include the additional results in revision.
>
> | Metric | DeepSeek-v3 | Claude 4 Sonnet | GPT-4o |
> |--------|---------------|-------------|-------------|
> | ASR    | 13.1              | 3.6       | 2.4       |
>
> Note that GPT-4o is not applicable to LARGO directly because we do not have access to the latent, which is needed as part of its optimization process. Nevertheless, the 4o provider, OpenAI, can use our framework to perform internal red-teaming to help expose the weakness of their own models and improve model safety.
>
> 4. **Benchmark against newer baselines**: Thank you for the suggestion. We already compared 3 recent jailbreak methods in our work. We add PAIR [3], a strong black-box baseline, and show results on Llama2-7b and Qwen2.5-14b below. PAIR underperforms on both models and takes significantly more LLM calls in its agentic prompting process. This further verifies the effectiveness of our LARGO.
>
> | Method | Llama2-7b | Qwen2.5-14b |
> |--------|-----------|-------------|
> | PAIR   | 2.5       | 17.5        |
> | LARGO  | 42.0      | 46.5        |
>
> Moreover, as highlighted by reviewer WCWm, “I have been working on a very similar research direction, so obviously I think this paper is a valuable contribution!’’, and they think “the formal/benchmark results are somewhat secondarily to the ‘it ain’t dumb if it works’. The main value of this paper is this simple and novel way to conduct the soft-to-hard rounding in a way that has non-trivial ASR on meaningful models!’’
>
> We believe that our innovative method connecting latent and discrete space is a meaningful contribution on its own. As the first paper doing this, the numbers are non-trivial too. We will include additional results in revision as suggested.
>
> **On Questions**
> 1. **Measurement of latent-only success**: We would like to clarify that the purpose of the reflection step is not to enhance jailbreak performance, but to convert powerful gradient-based attacks into benign text prompts. It is expected that latent only has a higher success rate, as it directly optimizes the latent space. LARGO’s goal is to search for text prompts in the input space that can jailbreak an LLM, which is a different setting with different application scenarios. Nevertheless, the LLM providers such as Google and OpenAI can use our framework to perform internal red-teaming to help expose the weakness of their own models, therefore developing safeguard accordingly.
>
> 2. **Joint optimization of latent representation and textual realization**: We are sorry for the confusion. Actually LARGO already runs a loop that jointly optimizes both the latent and textual realization. More details are provided Section 2.1.3 and in the pseudocode. We will make this point clearer in the revision as suggested.
>
> **On Limitations**
> 1. **White box dependency**: Though we require gradient access, yet for LLM providers, who have access to their own model gradient and everything, our algorithm can be used by them to perform red-teaming and find vulnerabilities.
>
> 2. **Self-reflection dependency**: Our method can be further improved as LLM’s capability on interpreting their own latent improves. As LLM develops, our method can be even more powerful in helping future LLM red teaming when future LLM gets better in interpreting themselves. Future work can even train LLM to self-reflect better.
>
> [1] SelfIE: Self-Interpretation of Large Language Model Embeddings, ICML 2024
>
> [2] Patchscopes: A Unifying Framework for Inspecting Hidden Representations of Language Models, ICML 2024
>
> [3] Jailbreaking Black Box Large Language Models in Twenty Queries, NeurIPS 2023

---

> > ### Comment · Reviewer_qRbx · 2025-08-04
> >
> > Thank you for the comprehensive and convincing additional experiments. The iterative optimization approach is indeed simple and provides useful insights; however, I believe it is still not sufficient to fully close the gap between LARGO and the latent-only variant. I hope authors can explore truly joint optimization strategies that more effectively bridge this gap. Based on the new results and thoughtful clarifications, I have increased my score to 4.

---

### Decision · Program_Chairs · 2025-09-17

**Decision:**

Accept (poster)

**Comment:**

This paper proposes LARGO - Latent Adversarial Reflection through Gradient Optimization, a novel jailbreak attack that optimizes an adversarial vector in the continuous latent space. It then recursively uses the same LLM to decode the latent into natural language. The method generates a fast, effective, and transferable attack that produces fluent and stealthy prompts.

The paper has the following strengths recognized by the reviewers:
- The idea of jailbreaking attack in the latent space is novel.
- The attack is fast and effective to generate transferable and fluent attacks.

There are few limitations of the paper:
- The evaluation is conducted on several outdated models: some of the latest models (e.g., Llama-3, Qwen2.5, or GPT-4o) are not evaluated. The authors have conducted further experiments on Qwen-2.5-14b and evaluate the transferability on GPT-4o/Claude 4. The authors are encouraged to further conduct experiments on more models to make the results more convincing.
- The transferability is limited: based on the new results, it seems that the transferability to commercial models is low. The authors are encouraged to discuss the results in the final version.
- The compared baselines are limited: the paper only compared with some old baselines. Adding the results of more recent baselines is necessary. The authors are encouraged to add more baseline in the final version.

After author rebuttal and discussions, all reviewers suggested to accept this paper. The paper provides novel contributions to the field by developing a jailbreaking attack in the latent space. However, the limitations listed above (mainly in evaluation) should be carefully considered in the final version. The AC would recommend acceptance.